# Factors Associated with Lack of Health Screening among People with Disabilities Using Andersen’s Behavioral Model

**DOI:** 10.3390/healthcare11050656

**Published:** 2023-02-23

**Authors:** Ye-Soon Kim, Seung Hee Ho

**Affiliations:** Department of Healthcare and Public Health Research, Rehabilitation Research Institute, Korea National Rehabilitation Center, Seoul 01022, Republic of Korea

**Keywords:** health screening, people with disabilities, Andersen’s behavior model

## Abstract

People with disabilities often have poorer health than the general population, and many do not participate in preventive care. This study aimed to identify the health screening participation rates of such individuals and investigate why they did not receive preventive medical services based on Andersen’s behavioral model, using data from the Survey on Handicapped Persons with Disabilities. The non-participation health screening rate for people with disabilities was 69.1%. Many did not in health screening because they showed no symptoms and were considered healthy, in addition to poor transportation service and economic limitations. The binary logistic regression result indicates that younger age, lower level education, and unmarried as predisposing characteristics; non-economic activity as the enabling resources; and no chronic diseases, severe disability grade, and suicidal ideation as need factor variables were the strongest determinants of non-participation health screening. This indicates that health screening of people with disabilities should be promoted while takings into account the large individual differences in socioeconomic status and disability characteristics. It is particularly necessary to prioritize ways to adjust need factors such as chronic disease and mental health management, rather than focusing on uncontrollable predisposing characteristics and enabling resources among barriers to participation in health screening for people with disabilities.

## 1. Introduction

Health screening aims to detect and treat diseases at an early stage, thereby reducing the burden of medical expenses and ensuring a healthy life [1]. In Korea, health screening services are divided into national and private health screenings, which differ in terms of screening items and cost burdens. National health screening mainly provides basic and essential health screening items, with little financial burden on individuals. In a private health screening, although various health screening items can be selected according to the individual’s characteristics and preferences, the economic burden is high because it is fully borne by the individual [2].

Korea’s national health screening aims to detect obesity, dyslipidemia, high blood pressure, and diabetes, which are risk factors for cardiovascular and cerebrovascular diseases, early and improve quality of life through treatment or lifestyle improvement. The Korean national health screening is aimed at checking health conditions and preventing and detecting diseases at an early stage. Health screening consists of examination and consultation, physical examination, diagnostic examination, pathology examination, radiological examination, etc., through health screening institutions [3,4].

The most representative health screening in Korea is that of the National Health Insurance Service. National health screenings have expanded in subjects and examination items since medical insurance health screening for public servants and teachers began in 1980. The national health screening participation rate in Korea in 2019 was 74.1% [5]. However, the health screening participation rate of people with disabilities was 64.6% [6]. Since the introduction of the national health screening, the increased rate of health screening participation and preparation strategies for health promotion shows its success. However, it was found that the health screening participation rate of people with disabilities was not only low, but this group also suffers from many chronic diseases [6]. Because of this, it is important to determine the cause of this reduced rate and take countermeasures. Although the rate of health screenings for people with disabilities is reported steadily, it is clear that there are deficiencies in implementing national policies and health promotion services for people with disabilities.

There are still no general or specialized health screening systems for people with disabilities to detect or prevent secondary diseases at an early stage. Article 7 of the Guarantee of the Right to Health and Medical Accessibility of Persons with Disabilities (Act on the Right to Health of Persons with Disabilities), enacted in December 2015 stipulates the “health screening project for persons with disabilities”; efforts were made at the national level to ensure customized health screening for people with disabilities [7]. Health screening items suitable for characteristics such as gender, sex, and life cycle should be designed. To do so means that it is necessary to identify the influencing factors related to the health screening of people with disabilities.

Previous studies related to health screening for people with disabilities have been reported by Park et al. [8], Yoon [9], Kim et al. [10], and the National Rehabilitation Center [11]. According to a study on the health screening rate of people with disabilities, screenings were lower among women with disabilities, those of an older age, and those receiving medical aid; the higher the income, the lower the health screening rate, and there are differences in the health screening participation depending on the type and grade of disability. In particular, it is reported that the screening rate decreases as the degree of disability increase from mild to severe and if the mobility disability is greater. A study in the United States also reported that the higher the degree of disability, the lower the screening rate for diseases such as cervical cancer [12]. In addition, the screening rate of people with disabilities is lower than that of the general population [13]. People with disabilities have the same rights to healthcare as the general population. To improve the health screening participation rate, which is also emphasized in The 5th Policy Plan for people with disabilities in South Korea [14], it is necessary to identify related factors. For this study’s purpose, health screening is also applied as part of medical utilization and Anderson’s behavioral model of health service utilization is applied. We looked at the actual health screening participation behavior and tried to predict the factors that caused this behavior.

Therefore, in this study, we tried to identify the status of health screening of people with disabilities and the factors affecting health screening by using the disability status survey, which provides sample statistical data for people with disabilities. The findings can help identify factors that affect the health screening of people with disabilities, as well as factors needed to improve the health screening rate. In addition, by identifying and addressing the factors influencing health screening by predisposing characteristics, enabling resources, and need factors, it is possible to grasp the current status of health screening for people with disabilities and re-examine it, providing evidence for follow-up tasks and research in the field of health for people with disabilities. This study aimed to examine the health screening rates of people with disabilities and the characteristics of those who did not undergo health screenings, and identify factors that affect health screening for people with disabilities. The specific research objectives were as follows: first, the sociodemographic characteristics of the people with disabilities were identified. Second, the general health screening rate of people with disabilities and reasons for not taking the examination were identified. Third, the characteristics of the predisposing characteristics, enabling resources, and need factors for general health screenings for people with disabilities and those who did not undergo health screenings were identified. Fourth, factors affecting general health screening of people with disabilities were analyzed.

## 2. Materials and Methods

This analytical study used the 2020 Survey of People with Disabilities, (as secondary data) to identify factors that affect the health screenings for people with disabilities based on Andersen’s behavioral model (Figure 1) [15]. Andersen’s behavioral model is a conceptual model aimed at demonstrating factors that lead to the use of health services. According to the model, usage of health services (including inpatient care, etc.) is determined by three dynamics: predisposing characteristics, enabling resources, and need factors. Predisposing characteristics can be factors such as sex, age, and health beliefs. Need factors represent both perceived and actual need for health care services. The original model was expanded through numerous iterations and its most recent form models past the use of services to end at health outcomes and includes health screening [16].

### 2.1. Participants and Analysis Data

This study used data from the 2020 Insolvency Survey conducted by the Ministry of Health and Welfare and the Korea Institute for Health and Social Affairs [17]. This is reflected in Korea’s Social Welfare Act, which has been renewed every three years since the 2007 legal system. The 2020 Survey on Handicapped Persons with Disabilities comprises data on contact disabilities obtained by surveying 11,210 registered persons across 248 survey areas in Korea. It is representative data that used two-stage cluster sampling considering type, degree of disability, and age of the target disabilities group. A total of 7025 people participated in this survey, of which 365 people under the age of 19 were excluded, and 6660 people were finally analyzed.

### 2.2. Analysis Variables

#### 2.2.1. Dependent Variable

Among the survey items for people with disabilities in 2020, based on the question “Have you had a health screening in the past two years (2018–2020)?” was used [17]. This survey included comprehensive health examinations paid for by the individual, special health examinations at industrial sites (for workers exposed to hazardous substances), health examinations from the National Health Insurance Service (for the workplace or regional subscribers and medical benefit recipients), and free health examinations (including health screening by local governments other than the National Health Insurance Corporation).

#### 2.2.2. Independent Variable

Predisposing factors

The predisposing factors included sociodemographic variables such as sex and age, and social structural variables such as occupation and education, which the individual already possesses, regardless of his or her will. Education level was divided into elementary school, middle school, high school, and university graduation. Marital status was divided into married (having a spouse) and other categories (single, widowed, divorced, separated, single mother/unmarried father, etc.).

Enabling resources

Enabling factors satisfy the need for medical services by enabling individuals to use medical services, such as income and medical security benefits. The enabling resources in this study were subjective economic house status, national health insurance, and economic activity. In the case of economic activity, “Did you work for income? “was identified through questions.

Need factors

Necessary factors are the pursuit of medical service because of the condition of the disease; in this study, the variables were disability type and grade, chronic disease, stress levels in daily life, feelings of sadness or despair, suicidal ideation, and suicide attempt. Concerning disability types, 15 categories were investigated in the survey: physical function disability, disability with a brain lesion, visual impairment, hearing impairment, speech impairment, intellectual disability, autistic disorder, mental disorder, kidney dysfunction, cardiac dysfunction, respiratory dysfunction, liver dysfunction, facial dysfunction, intestinal or urinary fistular, and epilepsy. However, these 15 disability types were adjusted to five considering the proportion: physical function disability, disability with a brain lesion, visual impairment, hearing impairment, and others considering the specific gravity. The ratings for each type of disability ranged from 1 to 6. Grade 1 refers to the most severe disability, while Grade 6 refers to the least severe disability. Usually, grades 1 to 3 represent people with severe disabilities, and grades 4 to 6 represent people with mild disabilities.

### 2.3. Data Analysis

We used SPSS Window 26.0 for data analysis, and the significance level was set at 0.05. The general and disability-related characteristics of people with disabilities were analyzed by frequency, percentage, mean, and standard deviation. The relationship between the predisposing characteristics, enabling resources, and need factors of the participants and the health examination for people with disabilities were verified using a chi-square test. To identify the factors that affect health screenings of people with disabilities, a multiple logistic regression analysis was performed, which included predisposing characteristics, enabling resources, and need factors as independent variables.

## 3. Results

### 3.1. General Characteristics

Regarding the general characteristics of the participants, 59.1% were male and 40.9% were female, with a male-to-female ratio of 6:4. Regarding age groups, 8.7% were aged 20–39 years, 28.8% were aged 40–59 years, 48.3% were aged 60–79 years, and 14.2% were aged 80 years or older. Regarding education level, 38.9% graduated from elementary school or less, 19.6% graduated from middle school, 36.2% graduated from high school, and 5.3% graduated from college or higher (including junior college). Regarding marital status, 50.7% were married and 49.3% were in “other”.

Regarding national health insurance, 71% were enrolled in health insurance, 27.1% in medical aid, and 1.8% in others. Regarding subjective house economic status, 70.2% of the participants belonged to “lower level”, 28.9% to the middle level, and 0.9% to the upper level, which showed that people with disabilities generally experience economic difficulties. Of the participants, 24.7% said they were engaged in economic activities, and 75.3% were not. Chronic diseases were present in 75.6% of the participants and absent in 24.4%. The disability types were physical function disability (26.6%), brain lesions (11.9%), vision impairment (11.7%), hearing impairment (14.6%), developmental issues (7.6%), and others (language, mental, and height problems; 27.6%). Disability grades were severe (grades 1–3; 49.4%) and mild (grades 4–6; 50.6%). The degree of stress in daily life was slight (14%), moderate (50.5%), and high (35.5%). Of the participants, 19.8%, 12.3%, and 0.7% people experienced sadness or hopelessness, suicidal thoughts, and suicide attempts, respectively; 80.2%, 87.7%, and 99.3% did not experience sadness or hopelessness, suicidal thought, and suicidal attempts, respectively (Table 1).

### 3.2. Health Screening Participation Rates and Reasons for Not Participation Health Screening

It was found that 69.1% of people with disabilities underwent health screening. The main reasons for not undergoing health screening were “lack of symptoms and being considered healthy” (32.9%), “convenience of transportation” (20.4%), “others reasons” (12.4%), “economic reasons” (8.2%), and “lack of time” (6.2%). In addition, there were opinions that responded: “Anxiety regarding health screening results”, “difficulty in communication”, “insufficient knowledge regarding health screening”, “insufficient facilities for people with disabilities in medical institutions”, “not having someone for the company when visiting a health screening institution.” There were also reasons such as “there is no reason” and “it is difficult to make a reservation for a screening institution” (Table 2).

### 3.3. Comparison of Factors According to Health Screening Status

There were significant differences in health screening rates related to age, education level, marital status, subjective house economic status, chronic diseases, health insurance, economic activity, disability type and grade, depressive symptoms, suicidal ideation, and suicide attempts. Regarding age groups, 60–80-year-old (52.8%) and 40–60-year-old (28.9%) participants showed higher health screening rates than those aged 80 (12.9%) and 20–40 years (5.4%). The age groups reported elsewhere were 20–39, 40–59, 60–79, ≥80 years. Elementary school graduates (37.7%) showed higher health screening rates than middle school (20.9%), high school (36.4%), or college (5.1%) graduates. Health screening rates were higher for those with spouses (56.6%) than those without a spouse (43.4%), and the health screening rate was high in the group with low subjective house economic status. Regarding the existence of national health insurance, the health insurance group (75.3%) had a higher health screening rate than those with medical aid (22.8%), and the non-economically active group (70.5%) had a higher screening rate than the economically active group (29.5%).

The health screening rate of those with chronic diseases (77%) was higher than that of the group without chronic diseases (23%), classified by disability type [physical disability (29.1%); brain lesion disorder (10.5%); visually impaired disability (12.9%); hearing impairment disability (15.6%)]. The screening rate for mild level (56.7%) was higher than that for severe level (43.3%) of people with disabilities. The health screening participation rate was high for people with disabilities that had relatively good mental health conditions, such as no depression (82.3%), no suicidal ideation (89.7%), and no suicide attempts (99.4%) (Table 3).

### 3.4. Analysis of Influencing Factors Related to Non-Participating in Health Screening

The results of the multi-logistic regression analysis on the nonparticipation of people with disabilities in health screening showed that age, education, marital status, type of medical insurance, economic activity, chronic diseases, degree of disability, and suicidal ideation were statistically significant at a significance level of 0.5 (Table 4). In terms of age, compared to those aged ≥80 years, the health screening rate in individuals in their twenties or thirties was approximately 2.1 times (95% CI = 1.4 to 2.9) lower. In terms of education, the probability of participation in health screening was 1.4 times lower for those with a lower education than for those with a higher education degree. The probability of not taking a health screening was approximately 1.3 times higher for people with disabilities without a spouse than for those with a spouse. Compared to national health insurance, the health screening participation rate of the medical aid group was approximately 1.2 times higher among those enrolled in health insurance schemes, and the rate of non-examination was twice as high among those who were not engaged in economic activities. Compared to those with physical disabilities, those with brain lesions and developmental disabilities were 1.6 times more likely to miss a health screening. The rate of non-examination for health screening was 1.4 times higher in cases of both no chronic diseases and severe disabilities. Those with suicidal ideation were 1.3 times more likely to fail health screening.

## 4. Discussion

Research on health screening rates for people with disabilities is often conducted sporadically. In this study, factors affecting the nonparticipation rate in health screening for people with disabilities were classified into predisposing characteristics, enabling resources, and need factors. The study aimed to provide basic data for establishing programs and policies that can improve the rate of health screenings for people with disabilities by analyzing the factors that affect non-participation in health screening for people with disabilities.

In this study, the health screening participation rate for adults with disabilities was 69.1%. Similar results were reported by Kim et al. which revealed a 70.2% health screening rate for people with disabilities [10]. In addition, the results of this study were 4.5% higher than the 64.6% health screening rate of people with disabilities in the 2019 health statistics for people with disabilities published by the National Rehabilitation Center [18], which reflected the results of the national health screening. Because this study included private health screenings in addition to national examinations, the results were higher than those of the National Rehabilitation Center. However, in 2019, the health screening rate for people without disabilities in Korea was 74% [18]. Therefore, the health screening participation rate of people with disabilities which was somewhat lower than that of the people without disabilities.

A study in the United States also reported that people with disabilities had lower screening rates than those without disabilities [13,19]. Few studies have quantitatively and qualitatively identified health screening rates of people with disabilities; therefore, comparison with existing studies is limited, making health screening an urgent task for people with disabilities. The first reason people with disabilities do not participate in health screening is that they have no other symptoms and think they are healthy. The prevalence of chronic diseases with disabilities is reported to be 86.4% [6]. Rather than waiting until the reason for visiting the hospital, it is necessary to detect and treat the disease early in an asymptomatic state and inform them of the need to improve their lifestyle. It has been found that uncomfortable transportation is a major barrier for people with disabilities, leading to non-participation in health screening. The government needs to establish a transportation system by expanding convenient mobility equipment in means of transportation, passenger facilities, and on the roads, and by improving the pedestrian environment, so that people with disabilities may travel safely and conveniently. In addition, a lack of information on health screenings, absence of guardians, and communication difficulties were found to be barriers to participation in health screenings for people with disabilities. For people with disabilities who have difficulty moving, policies such as ‘moving health screening service’ and ‘visiting health screening center’ are required for improvement.

In this study, the health of people with disabilities was analyzed according to age groups identified in previous studies [18,20], subjective economic status, economic activity, and degree of disability [8,20,21]. There was a difference in health screening participation rates. Although not significant in this study, there was a sex-based difference in the health screening rates of people with disabilities [21,22] Compared to men, women with disabilities had a lower health screening rate, meaning that their health is more vulnerable. In addition, a health screening strategy for people with low gross house income and severe disabilities is required. The results of the logistic regression analysis to understand the influence of variables that affect the health screening participation of people with disabilities showed that age group, subjective economic status, economic activity, and degree of disability had a statistically significant effect on the health screening rates. Older age, better subjective economic status, and milder symptoms were found to have a positive effect on the health screening participation rate. On the contrary, health screening rates were low for those with younger age, poor subjective economic status, and severe disabilities. In addition, of non-participation rate in health screening was 1.2 times higher for those without a spouse (unmarried, widowed, divorced, separated, single mother/unmarried father, etc.) than for those with a spouse.

This study has some limitations First, the survey data on the actual condition of people with disabilities depended on the participants’ responses to the question, “Have you had a health screening in the past two years?” In addition, it was not useful to segment and analyze various types of examinations, such as national general examinations, life transition period examinations, and cancer screening. Therefore, in the future, research identifying related factors with more diverse forms of examinations, such as health screenings during the transition period of life and cancer screenings, are required. Second, because the survey respondents were home-based people with disabilities, there could be limitations in representing all people with disabilities. Third, we cannot rule out that the critical variables of the factors affecting health screenings for people with disabilities are omitted because of the limiting variables. Various important variables, such as chronic disease status, region, and individual private insurance should be included. In this study, to increase the health screening participation rates for people with disabilities, age should be considered as a predisposing factor, economic level as an enabling factor, and severity of disability as a need factor. Based on these results, it is possible to improve the health screening rates of people with disabilities and establish health management and promotion policies to improve the health and happiness of people with disabilities, detect diseases early, and improve and promote current health conditions. Therefore, social and institutional support measures are required. In addition, appropriate rehabilitation services for people with disabilities are also required.

## 5. Conclusions

This study identified the factors affecting the health screening of 6660 people with disabilities aged 20 years or older who responded to the 2020 Survey on People with Disabilities. It is commonly known that people with disabilities have poor access to medical services compared to people without disabilities, considering their poor health and low economic status. Therefore, although the need for preventive medical services, such as health screening, is much higher for people with disabilities, its current provision is lower than that for people without disabilities. This inevitably leads to an increase in medical expenses [23,24]. Thus, the government requires active planning and design. Recently, the government invited people with disabilities to undergo health screening without any inconvenience, but the response rate was low. In general, for people with disabilities to receive health screening, facilities, equipment, and time must be customized. Accordingly, the government is building customized screening centers for people with disabilities. In addition to providing basic health screening services for people with disabilities through health screening centers, specialized health screening items should be developed and disseminated. Health promotion and disease prevention for people with disabilities can be achieved through the provision of customized health screening services for each life cycle considering the characteristics of people with disabilities, and more active and voluntary participation by the concerned people in the health screening to monitor their health at the national level. Considered that continuous efforts are also necessary to achieve a more suitable screening system for people with disabilities.

## Figures and Tables

**Figure 1 healthcare-11-00656-f001:**
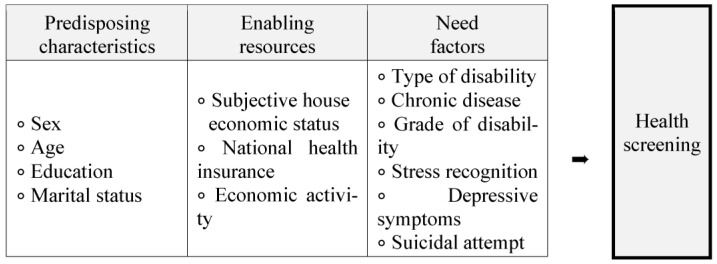
Study framework.

**Table 1 healthcare-11-00656-t001:** Characteristics of the subject of study (N = 6660).

Variables		N	(%)
Sex	Male	3935	−59.1
	Female	2725	−40.9
Age(years) (mean ± SD)		63.6 ± 15.1	
	20~39	579	−8.7
	40~59	1916	−28.8
	60~79	3217	−48.3
	≥80~	948	−14.2
Education	Uneducated/Elementary	2315	−38.9
	Middle school	1165	−19.6
	High school	2155	−36.2
	≥College	314	−5.3
Marital status	Married	3376	−50.7
	Others	3279	−49.3
Subjective house economic status	Low level	4676	−70.2
	Middle level	1926	−28.9
	High level	58	−0.9
National health insurance (NHI)	Health insurance	4731	−71
	Medical aid	1808	−27.1
	others	121	−1.8
Economic activity	No	5015	−75.3
	Yes	1645	−24.7
Type of disability	Physical function disability	1770	−26.6
	Disability of brain lesion	791	−11.9
	Visual impairment	782	−11.7
	Hearing impairment	975	−14.6
	Developmental disability	505	−7.6
	Others	1837	−27.6
Chronic disease	No	1622	−24.4
	Yes	5038	−75.6
Grade of disability	Severe (1~3 grade)	3290	−49.4
	Mild (4~6 grade)	3370	−50.6
Stress recognition	Little	935	−14
	Moderate	3361	−50.5
	High	2364	−35.5
Depressive symptoms	No	5343	−80.2
	Yes	1317	−19.8
Suicidal ideation	No	5840	−87.7
	Yes	820	(12.3)
Suicidal attempt	No	6611	−99.3
	Yes	49	−0.7

SD: standard deviation.

**Table 2 healthcare-11-00656-t002:** Participation rates and cause of non-participation in health screening experiences.

Variables		Health Screening				Total	
		Yes		No			
		N	(%)	N	(%)	N	(%)
Participation rates of untreated experiences							
	Untreated experiences	4599	−69.1	2061	−30.9	6660	−100
Cause of non-participation in health screening experiences							
	Lack of symptoms			678	−32.9		
	Poor transportation service			421	−20.4		
	Economic problems			170	−8.2		
	Lack of time			127	−6.2		
	Anxiety about the health screening results			90	−4.4		
	Communication problems			85	−4.1		
	Lack of knowledge about health screening			81	−3.9		
	Inadequate facilities for people with disabilities in medical institutions			68	−3.3		
	No one to accompany during visit			62	−3		
	Health screening problems			24	−1.2		
	Other			255	−12.4		

**Table 3 healthcare-11-00656-t003:** Comparison of factors according to participation and non-participation in health screening experiences (N = 6660).

Variables		Health Screening				Total		Chi-Square	
		Yes		No					
		N	(%)	N	(%)	N	(%)		
Sex									
	Male	2753	−59.9	1182	−57.4	3935	−59.1	3.709	
	Female	1846	−40.1	879	−42.6	2725	−40.9		
Age (years) (mean±SD)									
	20~39	247	−5.4	332	−16.1	579	−8.7	270.85	*
	40~59	1329	−28.9	587	−30.6	1916	−28.8		
	60~79	2430	−52.8	787	−38.2	3217	−48.3		
	≥80~	593	−12.9	355	−17.2	948	−14.2		
Education									
	Uneducated/Elementary	1530	−37.7	785	−41.6	2315	−38.9	17.359	*
	Middle school	849	−20.9	316	−16.8	1165	−19.6		
	High school	1477	−36.4	678	−35.9	2155	−36.2		
	≥College	207	−5.1	107	−5.7	314	−5.3		
Marital status									
	Married	2599	−56.6	777	−37.7	3376	−50.7	202.761	*
	Others	1995	−43.4	1284	−62.3	3279	−49.3		
Subjective house economic status									
	Low level	3091	−67.2	1585	−76.9	4676	−70.2	64.252	*
	Middle level	1462	−31.8	464	−22.5	1926	−28.9		
	High level	46	−1	12	−0.6	58	−0.9		
National health insurance (NHI)									
	Health insurance	3463	−75.3	1268	−61.5	4731	−71	142.829	*
	Medical aid	1048	−22.8	760	−36.9	1808	−27.1		
	Others	88	−1.9	33	−1.6	121	−1.8		
Economic activity									
	No	3242	−70.5	1773	−86	5015	−75.3	184.615	*
	Yes	1357	−29.5	288	−14	1645	−24.7		
Type of disability									
	Physical function disability	1338	−29.1	432	−21	1770	−26.6	288,636	*
	Brain lesion disability	484	−10.5	307	−14.9	791	−11.9		
	Visually impaired	592	−12.9	190	−9.2	782	−11.7		
	Hearing impairment	717	−15.6	258	−12.5	975	−14.6		
	Developmental disability	200	−4.3	305	−14.8	505	−7.6		
	Others	1268	−27.6	569	−27.6	569	−27.6		
	Chronic disease								
	Yes	3543	−77	1495	−72.5	5038	−75.6	15.65	*
	No	1056	−23	566	−27.5	1622	−24.4		
Grade of disability									
	Severe (1~3 grade)	1993	−43.3	1297	−62.9	3290	−49.4	218,618	*
	Mild (3~6 grade)	2606	−56.7	764	−37.1	3370	−50.6		
Stress recognition									
	Little	629	−13.7	306	−14.8	935	−14	55.415	*
	Moderate	2457	−53.4	904	−43.9	3361	−50.5		
	High	1513	−32.9	851	−41.3	2364	−35.5		
Depressive symptom									
	No	3787	−82.3	1556	−75.5	5343	−80.2	42.053	*
	Yes	812	−17.7	505	−24.5	1317	−19.8		
Suicidal ideation									
	No	4124	−89.7	1716	−83.3	5840	−87.7	54.182	*
	Yes	475	−10.3	345	−16.7	820	−12.3		
Suicidal attempt									
	No	4570	−99.4	2041	−99	6611	−99.3	2.25	
	Yes	29	−0.6	20	−1	49	−0.7		

* significantly different, *p* < 0.05.

**Table 4 healthcare-11-00656-t004:** Factors affecting health screening non-participation among people with disabilities (N = 6660).

Variables			OR †	95% CI ‡
Predisposing characteristics				
	Sex	Male	1	
		Female	1.075	(0.948~1.220)
	Age group	80 above	1	
		60~79	0.551	(0.491~0.658)
		40~59	791	(0.635~0.987)
		20~39 *	2.08	(1.491~2.902)
	Education	College or higher	1	
		High school	1.024	(0.775~1.427)
		Middle school	1.049	(0.775~1.354)
		Uneducated/Elementary *	1.409	(1.040~1.908)
	Marital status	Married	1	
		Others *	1.283	(1.124~1.466)
Enabling resources				
	Subjective house economic status	High level	1	
		Middle level *	0.889	(0.427~2.343)
		Low level	1.131	(0.546~1.851)
	National Health insurance (NHI)	Health insurance	1	
		Medical aid *	1.226	(1.061~1.417)
	Economic	Yes	1	
	activity	No *	2.078	(1.740~2.482)
Need factors				
	Type of disability	Physical disability	1	
		Disability of brain lesion *	1.606	(1.313~1.965)
		Visual impairment	0.899	(0.723~1.117)
		Hearing impairment	0.949	(0.779~1.157)
		Developmental disability *	1.605	(1.222~2.109)
		Others	1.142	(0.962~1.356)
	Chronic disease	Yes	1	
		No *	1.401	(1.204~1.630)
	Grade of disability	Mild (4~6 grade)	1	
		Severe (1~3 grade) *	1.54	(1.353~1.752)
	Stress recognition	Little	1	
		Moderate *	0.793	(0.664~0.947)
		High	1.096	(0.906~1.326)
	Depressive symptom	Yes	1	
		No	1.05	(0.881~1.252)
	Suicidal ideation	No	1	
		Yes *	1.324	(1.078~1.626)
	Suicidal attempt	No	1	
		Yes	1.552	(0.802~3.002)

* significantly different, *p* < 0.05. † OR: Odds ratio. ‡ CI: Confidence Interval.

## Data Availability

The database used in this study was provided by the Korea Institute for Health and Social Affairs (KIHASA), and is required by law to be distributed to the public free of charge. Therefore, the researcher accessed the KIHASA data-sharing service page “https://www.kihasa.re.kr (accessed on 13 December 2022)” and received data.

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
