# Peer review of "Factors Associated with Lack of Health Screening among People with Disabilities Using Andersen’s Behavioral Model"

_healthcare, 2023, doi:10.3390/healthcare11050656_

Round 1

Reviewer 1 Report

I had attempted to submit this earlier in the day, but I'm not sure that if it went through. Please find my review attached.

Author Response

Dear Reviewer,

  1. The authors note that socioeconomic status and disability characteristics are important to consider with health screening participation and that these findings could be used to inform health marketing or promotion activities (or future interventions). Based on the results, transportation appears to be an important barrier to screenings. Such findings could be beneficial for future screening centers and related efforts. Before being published, the article needs moderate revisions and additional editing. Suggested changes and edits are presented below.

The abstract contains unclear language which might be related to a find and replace error. The following section (copied below and in italics) should be deleted or thoroughly revised (with particular emphasis to the part in bold). I recommend keeping this as succinct as possible and focusing on the main findings and keeping the directionality of the findings the same throughout. For example, not participating in health screening was more likely among younger people, those with lower levels of education, unmarried individuals, people with less economic activity, those without a chronic disease, those with a severe disability, and those with suicidal ideation.

The probability of not undertaking the health screening participation was higher in the younger group than in the elderly group; lower level education group than in higher level education group; single group than in married group; those without who in economic activity than those with who in economic activity; those without who in chronic disease than those with who in chronic disease; those with a severe disability than those with a mild disability; and those with a suicidal ideation than those without a suicidal ideation.

  • Thank you for pointing this out. I have revised to the abstract for clearer language.

  1. In the introduction, more should be clarified about what happens during a health screening—what is the process like, how are screenings scheduled, how long does it take, etc.
  • Thank you for your suggestion. Accordingly, the following has been added to the introduction:

Korea’s national health screening aim to detect obesity, dyslipidemia, high blood pressure, and diabetes, which are risk factors for cardiovascular and cerebrovascular diseases, early and improve quality of life through treatment or lifestyle improvement. The Korean national health screening is aimed at checking health conditions and preventing and detecting diseases at an early stage. Health screening consists of examination and consultation, physical examination, diagnostic examination, pathology examination, radiological examination, etc., through health screening institutions.

  • References for newly added lines have also been added.

  1. In the materials and methods section, the Andersen behavioral model constructs (predisposing, possible, and necessary) are different than the figure with the study framework (predisposing, enabling, need), but I think they are supposed to align. These should be made consistent throughout or an explanation for why they are different is needed.
  • Thank you for pointing this out. The Andersen’s behavioral model components have been revised to match those in the Materials and Methods section.

  1. Regarding the analysis variables, I don’t think the dependent variable wording is right “Have disabled people undergone health screening for the past two years (2018–2020)?” Later in the manuscript it is written as “Have you had a health screening in the past two years?, which seems more likely to be correct.
  • Thank you. I have revised it to the latter sentence in the manuscript.

  1. The wording and coding for the independent variables should also be included. I was unsure what the difference between the health insurance and medical care group was. I was also unsure (and interested in) how economic activity was measured.
  • Thank you for your questions. The health security system in Korea has two components: mandatory social health insurance and medical aid. The National Health Insurance (NHI) system provides healthcare coverage to all citizens. The major sources of NHI funding include contributions from those who are insured and government subsidies. The medical aid program is a form of public assistance that uses government subsidies to provide low-income groups with healthcare services. Therefore, the terminology was revised to clarify meaning (medical insurance -> national health insurance (NHI); medical care -> medical aid)
  • An explanation of economic activity variables was also added. In the case of economic activity, "Did you work for income?” was identified through additional questions.

  1. In the results and throughout, the language for participating in a health screening or not participating in a health screening should be made more consistent. For example, my understanding was that failure to take a health examination (section 3.4) was the same as not participating in a health screening.
  • Thank you for pointing this out. I have revised the language to “participating in a health screening or not participating in a health screening.”

  1. 7. I recommend that the conclusion incorporates a few sentences on the importance of the social determinants of health (e.g., place, social position, income inequality, and access to opportunity) separate from health screening. This is important to the finding on transportation being a leading barrier to screening.
  • Thank you for this suggestion. I revised this section as requested.
  • Revised sentence: It has been found that uncomfortable transportation is a major barrier for people with disabilities, leading to non-participation in health screening. The government needs to establish transportation systems by expanding convenient mobility equipment in means of transportation, passenger facilities, and on the roads, and by improving the pedestrian environment, so that the people with disabilities may travel safely and conveniently. In addition, lack of information on health screenings, absence of guardians, and communication difficulties were found to be barriers to participation in health screenings for people with disabilities.

  1. The references should be made more consistent as well. Some dates are bolded (some are not). Specifically, the Anderson citation is in a different format from the rest (pasted below).
  2. Andersen R.A. 1968) Behavioral model of families' use of health services. University Of Chicago Research Series. 25, 25–32
  • Thank you for pointing this out. I have revised this item as requested.

  1. Figure 1: what is the p value? Are the differences observed statistically significant?
  • Figure 1 is the study framework. The results of the model in Figure 1 are presented in Table 4.

  1. Suggested line edits are presented below:
  • I have revised the language as requested.

Thank you.

Reviewer 2 Report

The article is interesting, reaching a subject important for social services: health for people with disabilities.

The abstract should be completed if included values of significance for each factor. The introduction is presenting the background of the problem studied and is backed up by references of studies published about this situation in Korea. There are also another articles that could be used for reference, like the one cited here: For references: Use of rehabilitation services by persons with disabilities in Brazil: A multivariate analysis from Andersen’s behavioral model Arthur de Almeida Medeiros , Maria Helena Rodrigues Galvão, Isabelle Ribeiro Barbosa, Angelo Giuseppe Roncalli da Costa Oliveira, Published: April 29, 2021, https://doi.org/10.1371/journal.pone.0250615 The materials and methods is reliable.

The results are well presented and discussion and conclusion clear.

I would recommend publication with minor changes, adding some other references on studied theme.

Author Response

Dear Reviewer,

  1. The abstract should be completed if included values of significance for each factor. The introduction is presenting the background of the problem studied and is backed up by references of studies published about this situation in Korea. There are also another articles that could be used for reference, like the one cited here: For references: Use of rehabilitation services by persons with disabilities in Brazil: A multivariate analysis from Andersen’s behavioral model Arthur de Almeida Medeiros , Maria Helena Rodrigues Galvão, Isabelle Ribeiro Barbosa, Angelo Giuseppe Roncalli da Costa Oliveira, Published: April 29, 2021,https://doi.org/10.1371/journal.pone.0250615 The materials and methods is reliable.

The results are well presented and discussion and conclusion clear.

  • Thank you for this suggestion. I have modified the discussion section using the recommended references. For the abstract, I have chosen to omit the specific values for each factor as this is explained in detail in the main text.

Thank you.

Reviewer 3 Report

Guided by Andersen's behavior model, this study examines the health screening rates of people with disabilities and identifies factors that influence health screening for people with disabilities. Using data from the Korean Survey on Handicapped Persons with Disabilities, they found the prevalence of non-examination participation rates was 69%. Demographic, socioeconomic, and health status are influential factors related to screening participation rates.

First and foremost, I recommend copy editing the current manuscript to increase the readability of your work. Especially, you may want to pay attention to grammar errors and broken sentences. Correcting those issues is essential to ensure that information is communicated clearly and efficiently.

I also suggest the authors mention the settings of this current research in the title. A brief description of the study setting should be added to the introduction.

In the introduction, a literature review on factors influencing screening participants rate should be included.  What are the factors we’ve known to influence screening participations among disabled patients?

Line 52-57 is loosely related to the research question. It should be explained what Article 7 is and what the other related policies are. A description of Andersen's behavior model should also be included in your theoretical model.

In terms of data, are the 11,210 registered people survey participants or those contacted for the survey? It is not clear how the survey was conducted.

How each variable is measured is not entirely clear; for instance, what are the categories for the occupation and how is health status assessed? The measurement should be described in great detail.

The discussion is a reiteration of the research findings. The authors should consider important questions on  how your findings reshape or add to existing knowledge about risk factors for non-participation in screening. Are there any policy suggestions we could make, etc.

Author Response

Dear Reviewer,

  1. Guided by Andersen's behavior model, this study examines the health screening rates of people with disabilities and identifies factors that influence health screening for people with disabilities. Using data from the Korean Survey on Handicapped Persons with Disabilities, they found the prevalence of non-examination participation rates was 69%. Demographic, socioeconomic, and health status are influential factors related to screening participation rates.

First and foremost, I recommend copy editing the current manuscript to increase the readability of your work. Especially, you may want to pay attention to grammar errors and broken sentences. Correcting those issues is essential to ensure that information is communicated clearly and efficiently.

  • Thank you for this suggestion. I have revised the language as requested. The overall manuscript was reviewed and revised.

  1. I also suggest the authors mention the settings of this current research in the title. A brief description of the study setting should be added to the introduction.
  • Thank you for this suggestion. I have added details to the introduction accordingly.
  • For the purpose of this study, health screening has also been applied as part of medical utilization, and Anderson's behavioral model of health service has also been used. We looked at the actual health screening participation behavior and tried to predict the factors that caused this behavior.

  1. In the introduction, a literature review on factors influencing screening participants rate should be included. What are the factors we’ve known to influence screening participations among disabled patients?
  • The factors are presented in the introduction:
    • According to a study on the health screening rate of people with disabilities, screenings were lower among women with disabilities, those of an older age, and those receiving medical aid; the higher the income, the lower the health screening rate, and there were differences in the health screening participation depending on the type and grade of disability.

  1. Line 52-57 is loosely related to the research question. It should be explained what Article 7 is and what the other related policies are. A description of Andersen's behavior model should also be included in your theoretical model.
  • Thank you for the suggestion. I have provided an explanation of what Article 7 is about:
    • Article 7 of the Guarantee of the Right to Health and Medical Accessibility of Persons with Disabilities (Act on the Right to Health of Persons with Disabilities), enacted in December 2015, stipulates the “health screening project for persons with disabilities”; efforts were made at the national level to ensure customized health screening for people with disabilities [7]. Health screening items suitable for characteristics such as gender, sex, and life cycle should be designed. To do so means that it is necessary to identify the influencing factors related to the health screening of the people with disabilities.
  • Andersen's behavioral model is presented in the materials and methods section as follows: Andersen’s behavioral model is a conceptual model aimed at demonstrating factors that lead to the use of health services. According to the model, usage of health services (including inpatient care) is determined by three dynamics: predisposing characteristics, enabling resources, and need factors. Predisposing characteristics can be factor such as sex, age, and health beliefs. Need factors represent both perceived and actual need for health care services. The original model was expanded through numerous iterations and its most recent form models past the use of services to end at health outcomes and includes health screening [16].
  1. In terms of data, are the 11,210 registered people survey participants or those contacted for the survey? It is not clear how the survey was conducted.
  • This survey collected information from 11,210 people with registered disabilities in 248 survey areas.
  • I have revised the text accordingly: The 2020 Survey on Handicapped Persons with Disabilities comprises data on contact disability obtained by surveying 11,210 registered persons across 248 survey areas in Korea.
  1. How each variable is measured is not entirely clear; for instance, what are the categories for the occupation and how is health status assessed? The measurement should be described in great detail.
  • Occupational variables have not been included in this study. Occupational variables were presented for the description of Andersen’s model. To eliminate confusion, the variable representation has been deleted.
  • Health status has also not been used as a variable in this study. I have deleted it because it was confirmed to be incorrectly marked. The chronic disease variable was originally excluded from study framework but was later added.
  1. The discussion is a reiteration of the research findings. The authors should consider important questions on how your findings reshape or add to existing knowledge about risk factors for non-participation in screening. Are there any policy suggestions we could make, etc.
  • It was confirmed that the factors for non-participation of people with disabilities in health screening were asymptomatic experience and traffic inconvenience. I have created a policy proposal for this.

For more details please see the revised version manuscript.

Thank you.

Round 2

Reviewer 3 Report

The revised version of this paper addresses most of the concerns I raised in the previous review round, and I appreciate the authors’ efforts to improve the manuscript. I have only one minor suggestion

Can the unit of measurement, type/coding of variable(s), etc used in the surveys be provided more clearly to illustrate the changes made by authors' choice of measures/coding/etc?

Author Response

Dear Reviewer,

  1. The revised version of this paper addresses most of the concerns I raised in the previous review round, and I appreciate the authors’ efforts to improve the manuscript. I have only one minor suggestion

Can the unit of measurement, type/coding of variable(s), etc used in the surveys be provided more clearly to illustrate the changes made by authors' choice of measures/coding/etc?

  • Thank you for this suggestion. We used 14 variables in this study (see Figure 1). The variables that require explanation were identified as type of disability and disability grade variables. Thus, we added the following: “Concerning disability types, 15 categories were investigated in the survey: physical function disability, disability with a brain lesion, visual impairment, hearing impairment, speech impairment, intellectual disability, autistic disorder, mental disorder, kidney dysfunction, cardiac dysfunction, respiratory dysfunction, liver dysfunction, facial dysfunction, intestinal or urinary fistular, and epilepsy. However, these 15 disability types were adjusted to five considering the proportion: physical function disability, disability with a brain lesion, visual impairment, hearing impairment, and others considering the specific gravity. The ratings for each type of disability ranged from 1 to 6. Grade 1 refers to the most severe disability, while Grade 6 refers to the least severe disability. Usually, grades 1 to 3 represent people with severe disabilities, and grades 4 to 6 represent people with mild disabilities.”

For more details please see the revised version manuscript. Thank you.